# Microarray Analysis for Transcriptomic Profiling of Myocardium in Patients with Fatal Myocardial Infarction

**DOI:** 10.3390/biomedicines11123294

**Published:** 2023-12-13

**Authors:** Vyacheslav Ryabov, Aleksandra Gombozhapova, Nikolai Litviakov, Marina Ibragimova, Matvey Tsyganov, Yulia Rogovskaya, Julia Kzhyshkowska

**Affiliations:** 1Cardiology Research Institute, Tomsk National Research Medical Center, Russian Academy of Sciences, 634012 Tomsk, Russia; rvvt@cardio-tomsk.ru; 2Cancer Research Institute, Tomsk National Research Medical Center, Russian Academy of Sciences, 634009 Tomsk, Russia; nvlitv72@yandex.ru (N.L.); imk1805@yandex.ru (M.I.); tsyganovmm@yandex.ru (M.T.); 3Tomsk Regional Oncology Dispensary, 634009 Tomsk, Russia; yuliya.rogovskaya@gmail.com; 4Institute of Transfusion Medicine and Immunology, Medical Faculty Mannheim, Heidelberg University, 69117 Heidelberg, Germany; julia.kzhyshkowska@googlemail.com; 5Laboratory of Translational and Cellular Biomedicine, National Research Tomsk State University, 634050 Tomsk, Russia

**Keywords:** myocardial infarction, gene expression, transcriptome, microarray analysis, TRAIL

## Abstract

Transcriptomic evidence from human myocardium in myocardial infarction (MI) is still not sufficient. Thus, there is a need for studies on human cardiac samples in relation to the clinical data of patients. The purpose of our pilot study was to investigate the transcriptomic profile of myocardium in the infarct zone, in comparison to the remote myocardium, in patients with fatal MI, via microarray analysis. This study included four patients with fatal MI type 1. We selected histologically verified samples from within the infarct area (n = 4) and remote myocardium (n = 4). The whole transcriptome was evaluated using microarray analysis. Differentially expressed genes (DEGs) clustered in the infarct area and in the remote myocardium allowed their differentiation. We identified a total of 1785 DEGs (8.32%) in the infarct area, including 1692 up-regulated (94.79%) and 93 down-regulated (5.21%) genes. The top 10 up-regulated genes were TRAIL, SUCLA2, NAE1, PDCL3, OSBPL5, FCGR2C, SELE, CEP63, ST3GAL3 and C4orf3. In the infarct area, we found up-regulation of seventeen apoptosis-related genes, eleven necroptosis-related, and six necrosis-related genes. Transcriptome profiling of the myocardium in patients with MI remains a relevant area of research for the formation of new scientific hypotheses and a potential way to increase the translational significance of studies into myocardial infarction.

## 1. Introduction

Myocardial infarction (MI) remains a leading cause of death throughout the world [1]. Reperfusion therapy, and percutaneous coronary intervention in particular, has led to a reduction in the acute infarct mortality [2]. However, there have been no further improvements in early survival during the last decade [3]. Furthermore, the mortality caused by post-infarction cardiac remodeling and consequent heart failure (HF) remains high [4,5].

The pathogenesis that underlies MI is not completely clear, nor is the pathophysiological heterogeneity of myocardial healing and remodeling. Infarct size is a major factor in the prognosis of patients with MI [6]. It results from a sequence of ischemia-induced and reperfusion-induced injuries [6,7]. Previously, MI was considered to be a manifestation of cardiomyocyte necrosis, which is a form of cell death that comprises the rupture of mitochondria and sarcolemma [7]. However, in the past twenty years, the involvement of more regulated forms of cardiomyocyte cell death in ischemia–reperfusion injury has been observed. These forms include apoptosis, necroptosis, and pyroptosis [8]. How and to what extent these forms of cell death contribute to the infarct size and interact with each other during ischemia–reperfusion injury is still not clear [9].

Insufficient knowledge in this area leads to the lack of effective, personalized approaches to the management of MI and post-infarction HF. Understanding the exact mechanisms of the response of the myocardium to ischemia-induced and reperfusion-induced injury, in their spatial context, could be the key to developing novel predictive, diagnostic, and therapeutic tools, including cardioprotection strategies [9,10].

Variations in gene expression underlie the wide range of physical, biochemical, and developmental differences seen among distinct cells and tissues. They may play a role in the difference between health and disorder, between stages of disease, or between spatial changes in organs and tissues during the pathological process.

During the last 30 years, technological development and approaches focusing on transcriptome analysis have been stimulated mostly by molecular cancer research [11]. One of the major reasons for success in this area has been the availability of relevant cell lines that can serve as reproducible sources of RNA, as well as the widespread accessibility of normal and pathological tissues. In contrast, most samples available to study pathophysiological processes underlying MI are limited to rodent myocardium and human peripheral blood. This is due to the difficulty of obtaining human myocardial samples, especially in the case of acute MI [10]. Various significant insights have been obtained via experimental models of MI, as well as investigation of blood components and biomarkers in patients with MI [12,13]. However, there are differences between the cardiac biology of animals and humans [14]. Furthermore, researchers investigating blood samples and conducting in vitro studies do not fully analyze all the compounds responsible for ischemic injury at the tissue level [15]. Thus, there is a need for studies on human cardiac samples in relation to the clinical data of patients. Their use in research projects could potentially increase the translational significance of the studies.

In our work, we suggest a research protocol based on using myocardium samples from patients who died due to acute MI type 1. The purpose of our pilot study presented herein was to investigate the transcriptome profile of myocardium in the infarct zone, in comparison to remote myocardium, in patients with fatal MI, via microarray analysis.

## 2. Materials and Methods

### 2.1. Clinical Data

The study included 4 patients (8 samples) with fatal MI type 1. All patients died within 48 h of MI. Brief characteristics of the patients are shown in Table 1.

### 2.2. Autopsy

We selected samples from within the infarct area and the remote myocardium for microarray analysis according to the results of the histopathological study. The volume of each sample was 60–70 mm^3^. The obtained tissue samples were stored in RNAlater solution (Ambion, Austin, TX, USA) at −80 °C as per the manufacturer’s instructions, until further analysis.

### 2.3. RNA Isolation

RNA from myocardium samples was isolated using the RNeasy Plus Mini Kit (Qiagen, Hilden, Germany) according to the manufacturer’s instructions. We used a NanoDrop-2000 spectrophotometer (Thermo Scientific, Waltham, MA, USA) to measure RNA concentration and quality. The concentration of RNA ranged from 80 to 250 ng/µL. The optical density ratios at 260/280 and 260/230 that were used to examine RNA quality were in the ranges of 1.95–2.05 and 1.90–2.31, respectively. The RNA Integrity Number (RIN) was assessed using a TapeStation instrument and an R6K Screen-Tape kit (Agilent Technologies, Santa Clara, CA, USA). The RIN values were 7.6–9.2.

### 2.4. Microarray Analysis

We evaluated the whole transcriptome in 4 patients using a Human Clariom S Assays microarray (Affymetrix, Santa Clara, CA, USA). Differentially expressed genes (DEGs) were determined in samples from the infarct area and remote myocardium. Sample preparation, hybridization, and scanning procedures were performed in accordance with the manufacturer’s protocol, using an Affymetrix GeneChip Scanner 3000 7G system (Affymetrix, USA).

### 2.5. DEG Identification and Statistics

Microarray analysis provided raw expression data. Transcriptome Analysis Console (TAC) software 4.0 was used to identify DEGs. Correction for multiple comparisons FDR (false discovery rate) was not used, due to the small number of samples. The DEG threshold was set to *p*-value <0.05 (fold change: >2 or <−2). ANOVA adjusted by eBayes was used to identify DEGs. The eBayes analysis corrected the variance of the ANOVA with an empirical Bayes approach that uses the information from all the probe sets to yield an improved estimate for the variance. A probe set was considered to be expressed if 50% of the samples in the dataset had values below the DABG (Detected Above Background) threshold. The DABG was set to 0.05 Pos/Neg AUC threshold >0.7.

## 3. Results

We studied the expression of 21,448 genes. A total of 1785 DEGs (8.32%), including 1692 up-regulated (94.79%) genes and 93 down-regulated (5.21%) genes in the infarct area, were identified (Figure 1a). DEGs clustered in the infarct area and in the remote myocardium allowed their differentiation (Figure 1b).

According to the fold change, the 10 most up-regulated genes were TNFSF10 (TRAIL), SUCLA2, NAE1, PDCL3, OSBPL5, FCGR2C, SELE, CEP63, ST3GAL3, and C4orf3 (Table 2). All of these genes were up-regulated in the infarct area, in comparison to the remote myocardium, while the expression levels of TNFSF10 (TRAIL), SUCLA2, and NAE1 were more than 10-fold higher in the infarct area. Brief summaries of the TNFSF10, SUCLA2, and NAE1 genes are presented in Table 3.

According to the *p*-value (<0.0001), the three most up-regulated genes were TRIQK, LRIG3, and FAM173B (Table 4). Brief summaries for these genes are shown in Table 5. Among these three genes, TRIQK is the least explored to date.

We identified pathways affected in response to myocardial ischemia. The 10 pathways most activated in the infarct area are presented in Table 6. Pathways were sorted by count. In addition, in the infarct area we revealed the up-regulation of seventeen apoptosis-related genes, eleven necroptosis-related genes, and six necrosis-related genes (Figure 2).

## 4. Discussion

In this study, we investigated the transcriptome profile of myocardium in the infarct zone, in comparison to remote myocardium, in patients with fatal MI, via microarray analysis. Microarray analysis is a high-throughput detection technology. It contains complementary DNA (cDNA) fragments that hybridize with specific RNA molecules in a sample [16]. Microarray analysis enables comparisons of the expression and regulation of thousands of genes, simultaneously [17]. During the last decade, the use of microarray technology to study gene expression has been surpassed by bulk RNA sequencing (RNA-seq) and single-cell RNA-seq (scRNA-seq), which are methods that provide a potentially more accurate quantification of the abundance of different transcripts and more individual information [11,17,18,19]. However, new technologies also confront various challenges. For instance, the workflow of RNA-seq analysis comprises the preparation of a sequencing library, including PCR amplification [20]. PCR amplification contributes to biases, which may extend to subsequent cycles [20]. Also, PCR amplifies various molecules with unequal probabilities, leading to the irregular amplification of cDNA molecules [21]. In terms of scRNA-seq, one of the main limitations is the technical noise of the resulting data [19,22]. This occurs due to the fact that material from one cell is insufficient for a comprehensive study, which further necessitates the generation of extra material during amplification. Another challenge is the cost of scRNA-seq experiments [19]. Thus, microarray analysis remains a technology that provides the efficiency and quality of detection, with a relatively low cost. For example, gene expression profiling with microarray analysis has enabled the prediction of prostate cancer [23] and increased inflammation in patients with sickle cell disease [24]. Furthermore, there is an increasing interest in identifying gene expression profiles using microarray technology in the diagnosis and risk prediction of MI [13].

In this study, we have proposed a research protocol based on using human myocardium samples from patients who died due to acute MI type 1. Most human studies focus on blood samples, taking into consideration the potential application of the identified DEGs as MI biomarkers [13,25,26]. Gene expression data from the human myocardium might be able to provide a different insight regarding the etiology, pathophysiology, and progression of the disease. However, such data are not readily available and are not likely to be used in routine clinical practice. Although endomyocardial biopsy (EMB) has assumed a fundamental place in the diagnostic work-up of a number of cardiac disorders, acute MI is not an indication for this invasive procedure [27]. Unfortunately, in patients with MI, the balance of risk to scientific benefit is not in favor of EMB. This is why experimental models of MI are an important source of myocardium samples for the research. Nevertheless, it is necessary to note that cross-species differences in genes and living environments are obvious obstacles to translating animal research to humans. Hence, there is a need for approaches to study human cardiac samples in relation to the clinical data of patients, to increase the translational significance of these studies.

It should be noted that despite the difficulties in obtaining myocardial tissue, the accumulation of data continues. For instance, in a recent study, a combination of scRNA-seq, chromatin accessibility, and spatially resolved transcriptomics was used to investigate the events of cardiac tissue reorganization and to characterize the cell-type-specific changes in gene regulation, providing an integrated spatial multi-omic map of cardiac remodeling after MI [10]. The authors profiled a total of 25 samples from 23 individuals, including non-transplanted donor hearts as controls (n = 4), as well as samples from tissues with the ischemic zone (n = 12), border (n = 3), and remote myocardium (n = 6) of patients with acute MI. These specimens were collected at different time points after the onset of clinical symptoms, before the patients received an artificial heart or a left-ventricular assist device because of cardiogenic shock and as a bridge to transplantation. The authors inferred the gene-regulatory networks differentiating cell states and projected this information onto specific tissue locations, thus mapping regulators controlling gene expression in different myocardial tissue zones and disease stages. The authors emphasized the importance of the spatial characterization of gene regulation in the human heart, in homeostasis and after MI. This fundamental work could serve as a reference for future single-cell genomics studies, with spatial gene expression data for the human heart. Nevertheless, the described method of obtaining myocardium samples is complex and available only in a small number of modern state-of-the-art clinics.

We identified the up-regulation of TRAIL (tumor necrosis factor (TNF)-related apoptosis-inducing ligand) in the infarcted myocardium of patients with MI. TRAIL is a pro-apoptotic transmembrane protein [28]. The membrane-bound TRAIL can be cleaved by cysteine proteases, leading to the formation of the soluble form of TRAIL. The most important sources of soluble TRAIL are monocytes and neutrophils [29]. There are five TRAIL receptors: death receptor 4 (DR4), death receptor 5 (DR5), decoy receptor 1 (DcR1), decoy receptor 2 (DcR2), and soluble decoy receptor or osteoprotegerin [30]. Decoy receptors bind TRAIL and keep it from binding to the death receptors, while the death receptors have domains necessary to induce apoptosis. Thus, TRAIL has both pro- and anti-apoptotic functions. TRAIL receptors are expressed in various tissues, including cardiomyocytes [31] and vascular smooth muscle cells [32]. Data from animal studies showed that TRAIL may contribute to the pathophysiology of cardiomyopathy [33], atherosclerosis [34], and abdominal aortic aneurysm [35]. Its role in MI and consequent HF is still not clear.

Studies on TRAIL in patients with MI comprised measurement of TRAIL concentrations and in vitro studies [36,37,38,39]. Secchiero et al. showed that in the acute phase of MI, serum concentration of TRAIL was decreased [36]. Its level did not significantly differ from the level in the control group after 6 to 12 months of follow-up. However, TRAIL predicted in-hospital, long-term mortality (12 months of follow-up) and the incidence of HF. Therefore, low TRAIL levels at discharge represent a possible predictor of future cardiovascular events following acute MI. In another case, Nakajima et al. compared the expression of TRAIL in peripheral blood mononuclear cells (PBMCs) from 26 patients in the acute phase of MI with that in PBMCs from 16 healthy control subjects, using flow cytometry and RT-qPCR [39]. Expression of TRAIL protein was significantly higher in patients with MI. These findings led to further studies of TRAIL and its receptors in the pathophysiology of MI and following complications.

Whether TRAIL plays a role in the death of cardiomyocytes during health or disease is still under investigation. In one of the recent studies, Wang et al. reported that the TRAIL pathway mediates ischemia–reperfusion injury in rats, pigs, and non-human primates; and that blockade of TRAIL can prevent myocardial cell death after MI [40]. In this experiment, the authors used RT-qPCR to confirm that DR5 is up-regulated in the heart during ischemia–reperfusion injury. The authors showed that TRAIL induced the death of cardiomyocytes and recruited and activated leukocytes, directly and indirectly causing cardiac injury. Furthermore, transcriptome profiling revealed increased expression of inflammatory cytokines in infarcted myocardium, which was reduced by TRAIL blockade. These findings indicate that TRAIL mediates MI directly, by targeting cardiomyocytes, as well as indirectly, by affecting myeloid cells, thereby reducing cardiomyocyte death and inflammation, respectively. Thus, blocking TRAIL represents a potential therapeutic strategy to treat MI.

TRAIL selectively induces apoptosis in transformed cells, but does not appear to kill normal cells. There is evidence that apoptosis, along with necrotic cell death, contributes to myocardial ischemia–reperfusion injury [41,42]. However, the relative contribution of apoptosis to the extent of cardiac damage remains controversial [8]. Several studies showed that the apoptotic component of cell death in the myocardium is triggered at the time of reperfusion and does not manifest during ischemia [43,44]. Other studies demonstrated the beginning of apoptosis after prolonged myocardial ischemia without reperfusion, or after a brief period of ischemia and following reperfusion [45,46]. A series of works challenged the role of apoptosis in ischemia–reperfusion injury [47,48]. This was explained by the low expression of proteins required for the apoptotic process in adult cardiomyocytes. Nonetheless, apoptosis of endothelial cells and leukocytes could affect cardiomyocyte survival and function [48], so anti-apoptotic therapeutic strategies may still be cardioprotective. In the present study, we found up-regulation of seventeen apoptosis-related genes in the infarct area, along with eleven necroptosis-related genes, and six necrosis-related genes (Figure 2). At the same time, some of the apoptosis signaling pathway genes that were activated in the infarct zone were also involved in necrosis and necroptosis signaling pathways (Figure 2). Only two out of six genes activated in the infarct zone were specific to necrosis, and seven of eleven genes were specific to necroptosis. Thus, our data indicate the contribution of apoptotic and necroptotic forms of cell death to myocardium injury in MI, rather than a necrotic form. These findings support the relevance of the contributions and interactions of various forms of cell death during ischemia–reperfusion injury in patients with MI. Also, our protocol could complement the current methodologies used to evaluate cell death in MI. However, distinguishing between different forms of cell death is still a challenging task because of their potential overlap, as well as the limitations of the assessment techniques [8]. Future investigations into pathways of cell death will be of interest for revealing their diverse effects on the myocardium, including the activation of an inflammatory response.

It is known that anti-inflammatory therapies targeting cytokines such as TNF-α, IL-6, and IL-1β may help to ameliorate ischemia–reperfusion injury [49,50]. Despite their effectiveness in animals, TNF blockers showed no clinical benefit in clinical trials on HF and MI, such as RENAISSANCE, RECOVER, and RENEWAL [51]. Undoubtedly, cross-species differences in genes and living conditions hinder the translation of experimental studies to clinical practice, as well as co-morbidities and the pharmacological backgrounds of patients. The proposed protocol may also allow us to study the influence of co-morbidities on the pathogenesis of MI, its complications, and outcomes. For instance, we may reveal the influence of such a common condition as prediabetes and the following inflammatory burden in pericoronary adipose tissue [52,53]. Thus, our protocol based on using myocardium samples from patients with fatal MI might become a modest but important addition to cardiac transcriptomics, biomedicine, and translational science.

### Study Limitations

A limitation of our study is the investigation of autopsy material. Undoubtedly, post mortem changes affect the results of any study, including transcriptome analysis. However, data on the extent of these effects in humans are limited, especially in the case of acute MI. In addition, the sample size in this study was small, since this was a pilot project and we firstly wanted to research the possibility of applying microarray analysis for transcriptome profiling of myocardium in patients with fatal MI.

## 5. Conclusions

Biomedical systems and translational medicine are rapidly developing areas of medical science, as a result of the use of high-throughput scientific technologies to overcome clinical challenges. Nevertheless, the number of biomarkers that allow the solving of clinical goals is not extensive. MI and HF are prime examples of this. Despite a large number of ongoing biomarker studies, there is no expected breakthrough. An example is the lengthy investigation of TRAIL and its role in MI. Measuring the intermediate step between genes and proteins—transcripts of mRNA—bridges the gap between the genetic code and the functional molecules that run cells [54].

Transcriptomic evidence from human myocardium in MI is still not sufficient. In many respects, this is due to the difficulty of obtaining human myocardial samples in the case of MI. Findings from our pilot study advance this goal, by providing transcriptomic profiling of myocardium in patients with fatal MI, via microarray analysis. We revealed that DEG analysis could clearly distinguish the infarct area and the remote myocardium. TRAIL was shown to be the most up-regulated gene in the infarct zone. Furthermore, our analysis showed that in the infarct area, the number of up-regulated apoptosis-related genes was higher than necroptosis-related and necrosis-related genes. Thus, transcriptome profiling of myocardium in patients with MI is a relevant area for the formation of new scientific hypotheses. The next important and large stage after screening is the investigation of local biochemical differences and functions at the protein level. In the case of MI, the investigation of gene expression, transcriptomic changes, and protein functions remains one of the main research areas in the search for new biomarkers and pathways to develop new diagnostic and therapeutic approaches.

## Figures and Tables

**Figure 1 biomedicines-11-03294-f001:**
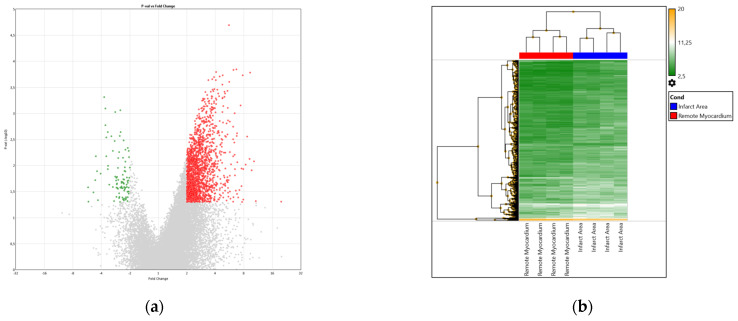
Panels: (**a**) Volcano plot of differentially expressed genes (DEGs) in the infarct area, in comparison to the remote myocardium. The results of microarray analysis. Up-regulated genes are marked in red, and down-regulated genes are marked in green. Transcriptome Analysis Console (TAC) software 4.0 was used to identify DEGs. The DEG threshold was set to *p*-value <0.05 (fold change: >2 or <−2). The ANOVA adjusted by eBayes was used to identify DEGs. (**b**) Heat map of differentially expressed genes (DEGs) in the infarct area and the remote myocardium. This figure shows that DEG analysis distinguishes the infarct area and the remote myocardium.

**Figure 2 biomedicines-11-03294-f002:**
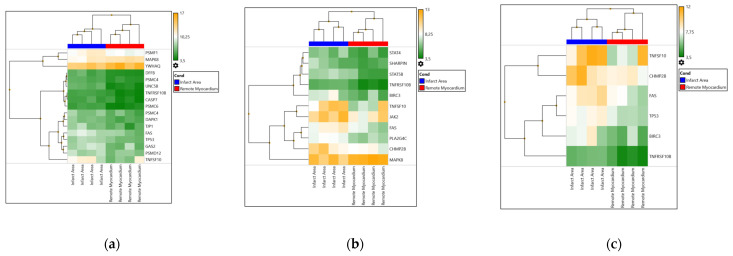
Heat map of differentially expressed genes (DEGs) associated with apoptosis (**a**), necroptosis (**b**), and necrosis (**c**) in the infarct area and the remote myocardium.

**Table 1 biomedicines-11-03294-t001:** Brief characteristics of patients.

Patient ID	1	2	3	4
Sex	Female	Male	Male	Female
Age	77	54	78	84
History of MI ^1^	-	-	-	+
GRACE score	12% (133 points)	13% (137 points)	40% (176 points)	20% (150 points)
PCI ^2^	PCI (stenting)	Primary PCI (balloon angioplasty)	Primary PCI (stenting)	PCI (balloon angioplasty)
Time from admission to death	48 h	90 min	3 h	5 h
Time from death to autopsy	10 h	21 h	10 h	15 h
Cause of death	Myocardial rupture	Cardiogenic shock	Cardiogenic shock	Cardiogenic shock

^1^ myocardial infarction; ^2^ percutaneous coronary intervention.

**Table 2 biomedicines-11-03294-t002:** The 10 most up-regulated genes in the infarct area, according to the fold change.

	Gene Symbol	Description	Fold Change	*p*-Val
1.	TNFSF10 (TRAIL)	Tumor necrosis factor (ligand) superfamily, member 10	19.83	0.049
2.	SUCLA2	Succinate-CoA ligase ADP-forming subunit beta	10.68	0.048
3.	NAE1	NEDD8 activating enzyme E1 subunit 1	10.24	0.008
4.	PDCL3	Phosducin like 3	9.65	0.012
5.	OSBPL5	Oxysterol binding protein like 5	9.32	<0.001
6.	FCGR2C	Fc gamma receptor IIc	9.15	0.008
7.	SELE	Selectin E	8.64	0.003
8.	CEP63	Centrosomal protein 63kDa	8.34	0.001
9.	ST3GAL3	ST3 beta-galactoside alpha-2,3-sialyltransferase 3	7.9	0.012
10.	C4orf3	Chromosome 4 open reading frame 3	7.87	0.045

**Table 3 biomedicines-11-03294-t003:** Brief summaries for the 3 most up-regulated genes in the infarct area, according to the fold change.

	Gene Symbol	Expression	Biological Processes
1.	TRAIL	Extracellular space Integral component of plasma membrane Extracellular exosome	Apoptotic process Activation of cysteine-type endopeptidase activity involved in apoptotic process (positive and negative regulation) Regulation of extrinsic apoptotic signaling pathway via death domain receptors Positive regulation of I-kappaB kinase/ NF-kappaB signaling Regulation of necrotic cell death Immune response Signal transduction Cell surface receptor signaling pathway Cell–cell signaling
2.	SUCLA2	Mitochondrion Mitochondrial matrix Myelin sheath Extracellular exosome	Tricarboxylic acid cycle Succinyl–CoA pathway Cellular metabolic process Small molecule metabolic process
3.	NAE1	Nucleus Cytoplasm Cytosol Plasma membrane	Protein binding NEDD8-activating enzyme activity Ubiquitin protein ligase binding Protein heterodimerization activity Small-protein-activating enzyme activity

**Table 4 biomedicines-11-03294-t004:** The 3 most up-regulated genes in the infarct area, according to the *p*-value.

	Gene Symbol	Description	Fold Change	*p*-Val
1.	TRIQK	Triple QxxK/R motif containing	<0.0001	5.58
2.	LRIG3	Leucine-rich repeats and immunoglobulin-like domains 3	<0.0001	6.68
3.	FAM173B (ATPSCKMT)	ATP synthase C subunit lysine N-methyltransferase	<0.0001	6.21

**Table 5 biomedicines-11-03294-t005:** Brief summaries for the 3 most up-regulated genes in the infarct area, according to the *p*-value.

	Gene Symbol	Expression	Biological Processes
1.	TRIQK	Endoplasmic reticulum membrane Integral component of membrane	May play a role in cell growth and maintenance of cell morphology
2.	LRIG3	Extracellular space Plasma membrane Cytoplasmic vesicle membrane	Otolith morphogenesis Multicellular organismal development Protein phosphorylation Transmembrane receptor protein tyrosine kinase signaling pathway Peptidyl–tyrosine phosphorylation
3.	FAM173B (ATPSCKMT)	Mitochondrial crista	Peptidyl–lysine trimethylation Positive regulation of sensory perception of pain Regulation of proton transport

**Table 6 biomedicines-11-03294-t006:** Top 10 pathways most affected in response to myocardial ischemia.

	Pathway	Number of Up-Regulated DEGs	Number of Down-Regulated DEGs
1.	miR-targeted genes in lymphocytes	40	1
2.	Protein–protein interactions in the podocyte	33	1
3.	PI3K–Akt signaling pathway	34	0
4.	VEGFA–VEGFR2 signaling pathway	32	2
5.	Focal adhesion PI3K–Akt–mTOR signaling pathway	33	0
6.	Nuclear receptors meta-pathway	29	1
7.	miR-targeted genes in muscle cell	29	1
8.	Malignant pleural mesothelioma	26	2
9.	MAPK signaling pathway	24	3
10.	Alzheimer’s disease	23	1

## Data Availability

The datasets used and/or analyzed during the current study are available from the corresponding author on reasonable request.

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
