# Peer review of "Microarray Analysis for Transcriptomic Profiling of Myocardium in Patients with Fatal Myocardial Infarction"

_biomedicines, 2023, doi:10.3390/biomedicines11123294_

Round 1
Reviewer 1 Report
Comments and Suggestions for Authors
The manuscript by Ryabov V et al. has shown the gene expression of MI patient's hearts using microarray technology. They report that apoptosis-related genes were overexpressed in thses hearts. They discussed the TRAIL gene and its significance very well in the study section.
Overall, I find the manuscript is well written and supported by the data presented.
Author Response
Dear Reviewer,
Thank you very much for your report and for appreciation of our work. We greatly value the time and effort you put into reviewing the manuscript. Thank you!
Reviewer 2 Report
Comments and Suggestions for Authors
This is a study investigating about the transcriptome profile of myocardium in the infarct zone in comparison to the remote myocardium in patients with fatal MI via microarray analysis. It is clear that biochemically different things are occurring between MI and non-MI areas, and I believe that biochemical considerations should also be considered when interpreting the transcriptome
There were other minor points.
# Conclusion was too long.
# I would like to know the difference between transcriptome in non-MI area and normal myocardium.
Comments on the Quality of English LanguageNo comment
Author Response
Dear Reviewer,
Thank you very much for your report. We greatly value the time and effort you put into reviewing the manuscript! We have revised the manuscript entitled «Microarray analysis for transcriptome profiling of myocardium in patients with fatal myocardial infarction» and would like to re-submit it for your consideration. We have addressed your valuable comments and we have conducted modification on the manuscript. The following are the point-by-point replies to the comments.
Question 1: This is a study investigating about the transcriptome profile of myocardium in the infarct zone in comparison to the remote myocardium in patients with fatal MI via microarray analysis. It is clear that biochemically different things are occurring between MI and non-MI areas, and I believe that biochemical considerations should also be considered when interpreting the transcriptome.
Reply: Thank you for this comment. Full-transcriptome analysis by both microarray and RNA-Seq are mainly screening methods that aim to identify a limited number of significant genes from a set of genes. These are the genes whose expression showed the greatest differences in two diametrically opposed myocardial conditions in the same patient. We also share your opinion that local biochemical differences at the level of proteins and their functions represent important and large stage of work. We are considering this phase as a next stage which follows after screening and that should be carried out separately, with all care. This statement is included in the conclusion (marked in green, page 10, line 320-322). Dear Reviewer, in case we misunderstood you, please, let us know.
Question 2: Conclusion was too long.
Reply: Thank you very much for your comment. We have corrected the conclusion (marked in yellow, page 10). Conclusion before correction includes 2574 characters with spaces. Conclusion after correction includes 1840 characters with spaces.
Question 3: I would like to know the difference between transcriptome in non-MI area and normal myocardium.
Reply: Thank you for your comment. The main highlight of our study was to compare paired samples with and without myocardial infarction from the same patient. We have assumed that comparison with normal myocardium may lead to high variability in many other features, which had an unpredictable effect on the result. Certainly, the difference between transcriptome in non-MI area and normal myocardium is still subject of scientific interest. Now that the first phase of work has been completed, we are considering a continuation of the study, including the study of the difference between transcriptome in non-MI area and normal myocardium.
Thank you again for your review. We believe that our work benefits from your expertise both now and in the future.
Round 2
Reviewer 2 Report
Comments and Suggestions for Authors
No comment
Comments on the Quality of English LanguageNo comment
Author Response
Dear Reviewer,
Thank you again for your report. We have revised the manuscript entitled «Microarray analysis for transcriptome profiling of myocardium in patients with fatal myocardial infarction» and would like to re-submit it for your consideration. We have edited English language and marked corrections in green.
Thank you.
